# Scalable Network Coding for Heterogeneous Devices over Embedded Fields

**DOI:** 10.3390/e24111510

**Published:** 2022-10-22

**Authors:** Hanqi Tang, Ruobin Zheng, Zongpeng Li, Keping Long, Qifu Sun

**Affiliations:** 1Department of Communication Engineering, University of Science and Technology Beijing, Beijing 100083, China; 2Network Technology Lab, Huawei Technologies Co., Ltd., Shenzhen 518000, China; 3Institute for Network Sciences and Cyberspace, Tsinghua University, Beijing 100084, China

**Keywords:** random linear network coding (RLNC), wireless broadcast network, scalable network coding

## Abstract

In complex network environments, there always exist heterogeneous devices with different computational powers. In this work, we propose a novel scalable random linear network coding (RLNC) framework based on embedded fields, so as to endow heterogeneous receivers with different decoding capabilities. In this framework, the source linearly combines the original packets over embedded fields based on a precoding matrix and then encodes the precoded packets over GF(2) before transmission to the network. After justifying the arithmetic compatibility over different finite fields in the encoding process, we derive a sufficient and necessary condition for decodability over different fields. Moreover, we theoretically study the construction of an optimal precoding matrix in terms of decodability. The numerical analysis in classical wireless broadcast networks illustrates that the proposed scalable RLNC not only guarantees a better decoding compatibility over different fields compared with classical RLNC over a single field, but also outperforms Fulcrum RLNC in terms of a better decoding performance over GF(2). Moreover, we take the sparsity of the received binary coding vector into consideration, and demonstrate that for a large enough batch size, this sparsity does not affect the completion delay performance much in a wireless broadcast network.

## 1. Introduction

In a communication network, linear network coding (LNC) advocates intermediate nodes to linearly combine received messages before transmission, so as to improve various network performances, such as increasing network throughput, reliability, and reducing transmission delay. Random linear network coding (RLNC) provides a distributed and asymptotically optimal approach for linear coding with coefficients randomly selected from a base field [1]. It shows the potential to improve the performance of unreliable or topologically unknown networks such as D2D networks [2], ad hoc networks [3], and wireless broadcast networks [4,5,6,7].

One of the reasons that hinder the large-scale practical applications of RLNC is the compatibility issue of different computational overheads. In complex network environments, there exist heterogeneous devices with different computational powers. Specifically, sources and certain receivers usually have ample computational powers while a large number of intermediate nodes and other receivers are computationally constrained such as the data collectors in ad hoc networks or low-cost devices in the Internet of Things paradigm [8]. It turns out that the coding compatibility among heterogeneous devices with different computational powers has to be considered in RLNC design.

This paper proposes a novel framework for scalable RLNC design based on embedded fields. The adjective *scalable* means that the finite fields chosen in the encoding process are not limited to a single base field but a set of *embedded fields* which consists of a large finite field and all its subfields. The encoding process at the source consists of two stages. In stage 1, based on a precoding matrix, all original packets are linearly combined over different finite fields to form precoded packets. In stage 2, the final packets to be transmitted are formed by randomly combining the precoded packets over GF(2). The heterogeneous receivers can recover the original packets over different fields under different computational constraints.

It is worthwhile to remark that prior to this work, there have been studies [9,10,11,12,13,14] that have taken different fields into account in the course of RLNC design. On one hand, the so-called Telescopic codes [9,10,11] and Revolving codes [12] considered different fields aiming at reducing the decoding complexity. However, they assume that all receivers have the same decoding capability, that is, they need support the arithmetic over the largest defined finite field. On the other hand, a flexible RLNC scheme called Fulcrum [13,14] makes use of GF(2) and its extension field GF($2^{8}$) for code design and it supports receivers to decode over both fields. Actually, Fulcrum can be regarded as a special instance in our proposed framework, while the decoding rule over GF(2) considered therein is weaker than the one proposed in this paper. In addition, there is limited discussion on the construction of an optimal encoding matrix in Fulcrum.

The main contributions of this paper are summarized as follows.

We mathematically justify how to make the arithmetic over different finite fields compatible.We derive a necessary and sufficient condition for decodability at a receiver over different finite fields. In particular, the proposed decoding rule over GF(2) is stronger than the one proposed in Fulcrum.We theoretically study the construction of an optimal precoding matrix in terms of the decodability performance.By numerical analysis in classical wireless broadcast networks, we demonstrate that the proposed scalable RLNC not only guarantees a better decoding compatibility over different fields compared with classical RLNC over a single field, but also provides a better decoding performance over GF(2) in terms of smaller average completion delay compared with Fulcrum.In numerical analysis, we also take the sparsity of the received binary coding vector into consideration, and demonstrate that for a large enough batch size, this sparsity does not affect the completion delay performance much in a wireless broadcast network.

This paper is structured as follows. Section 2 reviews the mathematical fundamentals of embedded fields. Section 3 first presents the general principles of the proposed scalable RLNC framework and then formulates the encoding and decoding process. Section 4 investigates the design of an optimal precoding matrix. Section 5 numerically analyzes the proposed scalable RLNC and compares its performance with classical RLNC over a single finite field as well as Fulcrum. Moreover, we take the sparsity into consideration and illustrate the influence on its performance. Conclusion is given in Section 6.

## 2. Mathematical Fundamentals

In our proposed scalable RLNC framework, different receivers will be able to recover the original packets over different finite fields, upon their different computational powers. In order to make the arithmetic over different finite fields compatible, we need the concept of embedded fields, which will be briefly reviewed in this section. One may refer to [15] for a detailed introduction on finite fields.

Recall that a finite field GF($2^{d_{1}}$) is a subfield of GF($2^{d_{2}}$) if and only if $d_{1}|d_{2}$. Thus, $\mathrm{GF}\left(2^{d_{1}}\right),\mathrm{GF}\left(2^{d_{2}}\right),\ldots ,\mathrm{GF}\left(2^{d_{D}}\right)$ are said to form embedded fields F if $d_{1}<d_{2}<\ldots <d_{D}$ and $d_{1}|d_{2}\left|\ldots \right|d_{D}$. For arbitrary GF($2^{d_{i}}$) and GF($2^{d_{j}}$) in F with i<j, as GF($2^{d_{j}}$) can be regarded as $\mathrm{GF}\left({\left(2^{d_{i}}\right)}^{d_{j}/d_{i}}\right)$, it can be expressed not only as a $d_{d_{j}}$-dimensional vector space over GF(2), but also as a $d_{j}/d_{i}$-dimensional vector space over GF($2^{d_{i}}$) at the same time.

**Example** **1.**
*Assume that $d_{1}=1$, $d_{2}=2$, $d_{3}=4$. The field GF($2^{4}$) can be expressed as a four-dimensional vector space over GF(2) as well as a two-dimensional vector space over GF($2^{2}$). Let α be a root of the irreducible polynomial $x^{2}+x+1$ over GF(2) so that GF$\left(2^{2}\right)=\left\{0,1,\alpha ,\alpha^{2}\right\}$. The polynomial $g\left(x\right)=x^{4}+x+1$ is irreducible over GF(2) but reducible over GF($2^{2}$) and can be factorized as $g\left(x\right)=\left(x^{2}+x+\alpha \right)\left(x^{2}+x+\alpha^{2}\right)$. Let β be a root of the irreducible polynomial $f\left(x\right)=x^{2}+x+\alpha$ over GF($2^{2}$) and β a root of f(x), so that $g\left(\beta \right)=\beta^{4}+\beta +1=0$ as well. Then, every element in $\mathrm{GF}(2^{4})$ can be expressed as $a_{0}+a_{1}\beta +a_{2}\beta^{2}+a_{3}\beta^{3}$ with $a_{j}\in \left\{0,1\right\}$. Moreover, $\alpha =\beta^{2}+\beta =\beta^{5}$, so that $\mathrm{GF}\left(2^{2}\right)=\left\{0,\beta^{0},\beta^{5},\beta^{10}\right\}$. Based on this, every element in GF($2^{4}$) can also be uniquely expressed as $b_{0}+b_{1}\beta$, $b_{0},b_{1}\in \mathrm{GF}\left(2^{2}\right)$, which is summarized in Figure 1. In Figure 1, the integers 0 to 15 are the decimal representation of the binary 4-tuple $\left(a_{3},a_{2},a_{1},a_{0}\right)$, e.g., 13 refers to $1+\beta^{2}+\beta^{3}$, which can be expressed 1+αβ.*


## 3. Framework Description

### 3.1. General Principles

In this paper, we focus on the construction of a general scalable RLNC framework over embedded fields, so we attempt to alleviate the influence of specific models of networks. In the course of framework description, we merely classify the nodes in a network into three types: a unique source node, intermediate nodes and receiver nodes. Assume that the source has the highest computational power, so that it can generate coded packets over embedded fields. The intermediate nodes in the network just recode the received data packets over GF(2), so as to fully reduce the overall computational complexities in the network. The heterogeneous receivers have different decoding capabilities. Under its own computational constraint, every receiver can judge whether sufficient coded packets have been received for decoding. More importantly, even though a receiver may not have sufficient computational power to deal with the arithmetic in a larger field over which some received packets are coded, it can still fully utilize these packets instead of directly throwing away in the process of decoding. For instance, assume that two received packets $w_{1}$ and $w_{2}$ are respectively equal to $p_{1}+p_{2}+\alpha p_{3}$ and $p_{2}+\alpha p_{3}$, where $p_{1},p_{2},p_{3}$ are original packets generated by the source node and α is an element not equal to 0 and 1 in the field GF($2^{d_{D}}$). For the receiver under the strongest field constraint GF(2), the original packet $p_{1}$ can be recovered by $w_{1}+w_{2}$ instead of directly throwing $w_{1}$, $w_{2}$ away. Consequently, the proposed scalable RLNC framework not only ensures the decoding capabilities of heterogeneous network devices but also fully reduces the required number of received packets for decoding.

### 3.2. Encoding and Recoding

In every batch, the source *s* has *n* original packets $p_{i},1\leq i\leq n$, each of which is an *M*-dimensional column vector over GF(2), to be transmitted to receivers. Without loss of generality, assume *M* is divisible by $2^{2^{D}}$, which can be achieved by padding dummy bits into every packet. With increasing *D*, the double exponentially increasing packet length *M* may cause the practical issue of an excessive padding overhead. Such an issue can be effectively solved based on the methods proposed in [16,17].

The encoding process at *s* has two stages. First, based on $p_{i},1\leq i\leq n$, for each 1≤d≤D, extra $r_{d}$ precoded packets are generated based on coding coefficients selected from GF($2^{2^{d}}$). In this process, every original packet $p_{i}$ is regarded as a vector of $m_{d}=M/2^{d}$ symbols, each of which consists of $2^{d}$ bits and represents an element in GF($2^{2^{d}}$). The multiplication of $p_{i}$ by a coefficient in GF($2^{2^{d}}$) is thus realized by symbol-wise multiplication. Note that when $d_{1}<d_{2}$, the coefficients in GF($2^{2^{d_{1}}}$) also appear in GF($2^{2^{d_{2}}}$), but the coding arithmetic changes. The mathematical fundamentals in the previous section guarantee the coding compatibility which will be illustrated in the next example.

**Example** **2.**
*Assume M=4, n=2, $d_{1}=1$ and $d_{2}=2$. Based on two original packets $p_{1}={\left[1\;0\;0\;0\right]}^{T}$ and $p_{2}={\left[1\;1\;0\;1\right]}^{T}$, a precoded packet is to be generated over $\mathrm{GF}\left(4\right)=\left\{0,1,\alpha ,\alpha^{2}\right\}$ by the linear combination $\alpha p_{1}+\alpha^{2}p_{2}$. First regard $p_{1}$ and $p_{2}$ as vectors of 2 symbols over GF($2^{2}$), that is, $p_{1}=\left[\begin{array}{c} \alpha \\ 0 \end{array}\right]$ and $p_{2}=\left[\begin{array}{c} \alpha +1\\ 1 \end{array}\right]=\left[\begin{array}{c} \alpha^{2}\\ 1 \end{array}\right]$. Then,*

(1)
$$\alpha p_{1}+\alpha^{2}p_{2}=\left[\begin{array}{c} \alpha^{2}\\ 0 \end{array}\right]+\left[\begin{array}{c} \alpha \\ \alpha^{2} \end{array}\right]=\left[\begin{array}{c} 1\\ \alpha +1 \end{array}\right]={\left[0\;1\;1\;1\right]}^{T}$$

*According to Figure 1, in GF($2^{4}$), $\alpha =\beta^{2}+\beta =\beta^{5}$ and $\alpha^{2}=\beta^{2}+\beta +1=\beta^{10}$. As every element in GF($2^{4}$) = GF($4^{2}$) can be uniquely expressed as $b_{0}+b_{1}\beta$, $b_{0},b_{1}\in \mathrm{GF}\left(4\right)$, every four-dimensional vector ${\left[a_{3}\;a_{2}\;a_{1}\;a_{0}\right]}^{T}$ over GF(2) as the following element in GF(16)*

$${\left[a_{3}\;a_{2}\;a_{1}\;a_{0}\right]}^{T}=a_{3}\beta^{6}+a_{2}\beta +a_{1}\beta^{5}+a_{0}.$$

*Based on this rule, $p_{1}=\beta^{6}$ and $p_{2}=\beta^{6}+\beta +1$. Consequently, $\beta^{5}p_{1}+\beta^{10}p_{2}=\beta +\beta^{10}=\beta +\beta^{5}+1$, which is ${\left[1\;\alpha +1\right]}^{T}$ over GF(4) and ${\left[0\;1\;1\;1\right]}^{T}$ over GF(2), same as (Equation 1) obtained by the GF(4) arithmetic.*


After stage 1, there are a total of $N=n+r_{1}+r_{2}+\ldots +r_{D}$ precoded packets, the first *n* of which are just the original packets. Let $G=\left[\begin{array}{c} I_{n}\;A_{1}\;\ldots \;A_{D} \end{array}\right]$ denote the n×N precoding matrix for the *N* precoded packets, where $I_{n}$ refers to the n×n identity matrix and $A_{d}$ is a coefficient matrix defined over GF($2^{2^{d}}$).

In stage 2, every coded packet c the source finally sends out is a random GF(2)-linear combination of the *N* precoded packets, that is,
$$c=[p_{1}\;p_{2}\;\cdots \;p_{n}]Gh,$$
for some randomly generated *N*-dimensional column vector h over GF(2), which is referred to as the *coding vector* for packet c. For a *systematic* scheme, the first *n* coded packets $c_{1},\ldots ,c_{n}$ transmitted by the source are just *n* original packets, that is, the coding vector for $c_{j}$ is just an *N*-dimensional unit vector with the $j^{\mathrm{th}}$ position equal to 1. Every coded packet will affix its coding vector to its header. In contrast, the information of precoding matrix G can either be affixed to the header of every packet or presettled to be known at every receiver.

At an intermediate node, the coded packets it transmits are GF(2)-linear combinations of its received packets. Specifically, if an intermediate node receives coded packets $c_{1},\ldots ,c_{l}$ with respective coding vectors $h_{1},\ldots ,h_{l}$, then it will recode them to generate a new coded packet $c^{\prime }$ to be transmitted as
$$c^{\prime }=a_{1}c_{1}+\ldots +a_{l}c_{l},$$
where $a_{1},\ldots ,a_{l}$ are random binary coefficients. The concomitant coding vector for $c^{\prime }$ is $a_{1}h_{1}+\ldots +a_{l}h_{l}$.

It is worthwhile to note that prior to this work, a flexible RLNC scheme called Fulcrum has been investigated in [13,14]. Fulcrum can be regarded as a special instance in our proposed framework with the setting D=3 and $r_{1}=r_{2}=0$.

### 3.3. Decoding

Define a linear map $\varphi :\mathrm{GF}{\left(2\right)}^{N}\to \mathrm{GF}{\left(2^{2^{D}}\right)}^{n}$ by
φ(v)=Gv.for every column vector $v\in \mathrm{GF}{\left(2\right)}^{N}$. The notation φ also applies to a set V of vectors: φ(V)={φ(v):v∈V}.

Moreover, let $U_{d}$, 0≤d≤D, denote the vector subspace of $\mathrm{GF}{\left(2\right)}^{N}$ spanned by unit vectors $u_{1},u_{2},\ldots ,u_{\sum_{d^{\prime }=0}^{d}r_{d^{\prime }}}$ where a unit vector $u_{j}$ refers to an *N*-dimensional vector with the only nonzero entry at position *j*.

For a receiver *t*, assume $d_{t}$ is the largest field for computation, and *m* packets have been received. Let H denote the N×m matrix over GF(2) obtained by columnwise juxtaposition of the coding vectors of the *m* received packets, and H the column space (over GF(2)) of H.

In order to recover original packets under the field constraint GF($2^{2^{d_{t}}}$), we need make use of coding packets with coding vectors in $U_{d_{t}}\cap H$ rather than in H. This is because the lower $\sum_{d^{\prime }>d_{t}}r_{d^{\prime }}$ entries in every coding vector corresponds to the original precoded packets generated by the source over a larger field than GF($2^{2^{d_{t}}}$). We next characterize the following necessary and sufficient condition for decodability at *t* up to field constraint GF($2^{2^{d_{t}}}$).

**Theorem** **1.**
*Based on the m received packets, the original n source packets can be recovered at t if and only if*

(2)
$$\dim(\varphi \left(U_{d_{t}}\cap H\right))=n.$$



**Proof.** First assume (Equation 2) holds. Then, there must exist n vectors, denoted by $v_{1},\ldots ,v_{n}$ in $U_{d_{t}}\cap H$ such that
(3)$$\dim(\varphi \left(\left\{v_{1},\ldots ,v_{n}\right\}\right))=n.$$Consequently, there exists an m×n matrix K over GF(2) such that $[v_{1}\;\ldots \;v_{n}]=HK$, and (Equation 3) implies the full rank *n* of GHK. As the last $\sum_{d^{\prime }>d_{t}}r_{d^{\prime }}$ rows in HK are all zero, the elements in GHK belong to GF($2^{2^{d_{t}}}$), and hence there exists an n×n matrix D over GF($2^{2^{d_{t}}}$) subject to $GHKD=I_{n}$, that is, the original packets can be recovered at *t*.Next assume that the original *n* packets can be recovered at *t*. Then, there exists an m×n matrix D over GF($2^{2^{d_{t}}}$) such that $GHD=I_{n}$. Further, D can be written as $D_{1}D_{2}$, where $D_{1}$, $D_{2}$ are over GF($2^{2^{d_{t}}}$) and of respective size m×n and n×n. Thus, $GHD_{1}$ is a matrix over GF($2^{2^{d_{t}}}$) of full rank *n*. Recall that none of the elements in the last $\sum_{d^{\prime }>d_{t}}r_{d^{\prime }}$ columns in G is in GF($2^{2^{d_{t}}}$). Thus, every element in $GHD_{1}$ belonging to GF($2^{2^{d_{t}}}$) implies that the last $\sum_{d^{\prime }>d_{t}}r_{d^{\prime }}$ rows in $HD_{1}$ are all zero. Moreover, as H is defined over GF(2), we can further deduce that $D_{1}$ can be written as $D_{1}^{\prime }D_{1}^{\prime \prime }$ for an m×n matrix $D_{1}^{\prime }$ over GF(2) and an n×n matrix $D_{1}^{\prime \prime }$ over GF($2^{2^{d_{t}}}$), such that the last $\sum_{d^{\prime }>d_{t}}r_{d^{\prime }}$ rows in $HD_{1}^{\prime }$ are all zero too, that is, the columns in $HD_{1}^{\prime }$ belong to $U_{d_{t}}\cap H$. In addition, the full rank of $GHD_{1}$ implies the full rank of $GHD_{1}^{\prime }$. Equation (Equation 2) is thus proved to hold. ☐

Based on the above theorem, we can further characterize the following equivalent condition for decodability at a receiver from the perspective of matrix rank. For 0≤d≤D, denote by $H_{d_{t}}$ the $\sum_{d^{\prime }>d_{t}}r_{d^{\prime }}\times m$ submatrix of H obtained by restricting H to the last $\sum_{d^{\prime }>d_{t}}r_{d^{\prime }}$ rows.

**Corollary** **1.**
*Based on the m received packets, the original n source packets can be recovered at t if and only if*

(4)
$$\mathrm{rank}(G\left(HK_{d_{t}}\right))=n,$$

*where $K_{d_{t}}$ is an $m\times (m-\mathrm{rank}\left(H_{d_{t}}\right))$ matrix whose columns constitute a basis for the kernel of the column space of $H_{d_{t}}$ such that $H_{d_{t}}K_{d_{t}}=0$.*


Note that the column space of $HK_{d_{t}}$ are exactly the subspace $U_{d_{t}}\cap H$ in (Equation 2), and all entries in the last $\sum_{d^{\prime }>d_{t}}r_{d^{\prime }}$ rows of $HK_{d_{t}}$ are zero, so the computation of (Equation 4) only involve arithmetic over GF($2^{2^{d_{t}}}$). Moreover, in order to check (Equation 4), it suffices to select $\mathrm{rank}(HK_{d_{t}})$ linearly independent column vectors in $HK_{d_{t}}$, juxtapose them into a matrix $H^{\prime }$, and check whether $\mathrm{rank}(GH^{\prime })=n$. With the number *m* of received packets at *t* increasing, the matrix $K_{d_{t}}$ and $H^{\prime }$ can be established in the following iterative way.

**Algorithm** **1.***Denote by $h^{m}$ the *N*-dimensional coding vector over GF(2) for the $m^{th}$ received packet at receiver t. Without loss of generality, assume that there is at least one non-zero entry in $h^{m}$. Let $h_{d_{t}}^{m}$ denote the vector restricted from $h^{m}$ to the last $\sum_{d^{\prime }>d_{t}}r_{d^{\prime }}$ entries. The next procedure efficiently produces desired $K_{d_{t}}$ and $H^{\prime }$*.*Initialization. Let $K_{d_{t}}$, $H^{\prime }$, B and $B_{d_{t}}$ be empty matrices. They are to consist of a *m* rows, N rows, N rows and $\sum_{d^{\prime }>d_{t}}r_{d^{\prime }}$ rows respectively*.*Iteration. Consider the case that the $m^{th}$ packet with coding vector $h^{m}$ is just received, and assume receiver t has dealt with the former m−1 coding vectors $h^{j},1\leq j<m$. Perform either of the following two depending on $h_{d_{t}}^{m}$*.
*If $h_{d_{t}}^{m}$ is a zero vector, then update $K_{d_{t}}$ as*(5)$$K_{d_{t}}=\left[\begin{array}{cc} K_{d_{t}} & 0\\ 0\ldots 0 & 1 \end{array}\right],$$*and respectively append a zero column vector to B and to $B_{d_{t}}$ on the right. Further check whether $h^{m}$ is a GF(2)-linear combination of columns in $H^{\prime }$. If so, keep $H^{\prime }$ unchanged. Otherwise, update $H^{\prime }$ as $\left[H^{\prime }\;h^{m}\right]$. The iteration for the current value of *m* completes*.*If $h_{d_{t}}^{m}$ is not a zero vector, check whether it is a GF(2)-linear combination of columns in $B_{d_{t}}$. If no, respectively update B, $B_{d_{t}}$ and $K_{d_{t}}$ as*(6)$$B=\left[B\;h^{m}\right],B_{d_{t}}=\left[B_{d_{t}}\;h_{d_{t}}^{m}\right],K_{d_{t}}=\left[\begin{array}{c} K_{d_{t}}\\ 0\ldots 0 \end{array}\right],$$*and the iteration for the current value of m completes. Otherwise, perform the following steps. First compute an (m−1)-dimensional vector k subject to $B_{d_{t}}k=h_{d_{t}}^{m}$, and then update $K_{d_{t}}$ as*(7)$$K_{d_{t}}=\left[\begin{array}{cc} K_{d_{t}} & k\\ 0\ldots 0 & 1 \end{array}\right].$$*Further compute a new vector $v=Bk+h^{m}$, and respectively append a zero column vector to B and to $B_{d_{t}}$ on the right. Check whether v is a GF(2)-linear combination of columns in $H^{\prime }$. If so, keep $H^{\prime }$ unchanged. Otherwise, update $H^{\prime }$ as $\left[H^{\prime }\;v\right]$. The iteration for the current value of *m* completes*.*Note that after the above procedure, the sum of the number of nonzero columns in $B_{d_{t}}$ and the number of columns in $K_{d_{t}}$ is m. The nonzero columns of $B_{d_{t}}$ keep to form a basis of the column space of $H_{d_{t}}=\left[h_{d_{t}}^{1}\ldots h_{d_{t}}^{m}\right]$. The columns of $K_{d_{t}}$ keep to form a basis of the null space spanned by columns of $H_{d_{t}}$. The columns in $H^{\prime }$ keep to be a basis of the column space of $HK_{d_{t}}$, where $H=[h^{1}\ldots h^{m}]$.*

**Example** **3.**
*Assume that D=2, $n=r_{0}=3$, and $r_{1}=r_{2}=1$. The 3×5 precoding matrix G is designed as*

$$G=\left[\begin{array}{ccccc} 1 & 0 & 0 & \alpha & \beta \\ 0 & 1 & 0 & \alpha^{2} & \beta \\ 0 & 0 & 1 & 1 & \beta \end{array}\right]$$

*where β is a primitive element in GF($2^{4}$) and $\alpha =\beta^{5}$, which can be regarded as a primitive element of $\mathrm{GF}\left(2^{2}\right)\subset \mathrm{GF}\left(2^{4}\right)$.*

*Assume that at a receiver t, GF($2^{2}$) is the largest field for computation, and 4 packets have been received with the columnwise juxtaposition of the respective coding vectors prescribed by*

$$H=\left[\begin{array}{cccc} 1 & 0 & 0 & 1\\ 0 & 1 & 0 & 1\\ 0 & 0 & 1 & 1\\ 0 & 0 & 0 & 1\\ 1 & 1 & 1 & 1 \end{array}\right]$$

*As $H_{d_{t}}=\left[1\;1\;1\;1\right]$ herein, the aforementioned iterative approach can yield the following $K_{d_{t}}$ and concomitant $H^{\prime }$:*

$$K_{d_{t}}=\left[\begin{array}{ccc} 1 & 1 & 1\\ 1 & 0 & 0\\ 0 & 1 & 0\\ 0 & 0 & 1 \end{array}\right],H^{\prime }=\left[\begin{array}{ccc} 1 & 1 & 0\\ 1 & 0 & 1\\ 0 & 1 & 1\\ 0 & 0 & 1\\ 0 & 0 & 0 \end{array}\right],$$

*where the columns of $H^{\prime }$ form a basis for the subspace $U_{d_{t}}\cap H$. Consequently, $GH^{\prime }=\left[\begin{array}{ccc} 1 & 1 & \alpha \\ 1 & 0 & 1+\alpha^{2}\\ 0 & 1 & 0 \end{array}\right]$. Since $1+\alpha +\alpha^{2}=0$ in GF($2^{2}$), $\mathrm{rank}(GH^{\prime })=2$, that is, (Equation 4) does not hold. Therefore, the receiver requires to receive more packets before decoding all original packets.*

*Assume $h^{5}={\left[1\;0\;0\;1\;1\right]}^{T}$ is the coding vector for the $5^{\mathrm{th}}$ received packet. Then, the matrix $K_{d_{t}}$ is dynamically updated to*

$$K_{d_{t}}=\left[\begin{array}{cccc} 1 & 1 & 1 & 1\\ 1 & 0 & 0 & 0\\ 0 & 1 & 0 & 0\\ 0 & 0 & 1 & 0\\ 0 & 0 & 0 & 1 \end{array}\right],$$

*but there is no change for $H^{\prime }$, because $H\cdot {\left[1\;0\;0\;0\;1\right]}^{T}$ belongs to the column space of $H^{\prime }$.*

*Assume $h^{6}={\left[0\;0\;1\;0\;0\right]}^{T}$ is the coding vector for the $6^{\mathrm{th}}$ received packet. First, dynamically update $K_{d_{t}}$ to*

$$\begin{array}{c} K_{d_{t}}=\left[\begin{array}{ccccc} 1 & 1 & 1 & 1 & 0\\ 1 & 0 & 0 & 0 & 0\\ 0 & 1 & 0 & 0 & 0\\ 0 & 0 & 1 & 0 & 0\\ 0 & 0 & 0 & 1 & 0\\ 0 & 0 & 0 & 0 & 1 \end{array}\right], \end{array}$$

*Then, as $h^{6}={\left[0\;0\;1\;0\;0\right]}^{T}$ does not belong to the column space of $H^{\prime }$, update $H^{\prime }$ as $\left[H^{\prime }\;h^{6}\right]$:*

$$H^{\prime }=\left[\begin{array}{cccc} 1 & 1 & 0 & 0\\ 1 & 0 & 1 & 0\\ 0 & 1 & 1 & 1\\ 0 & 0 & 1 & 0\\ 0 & 0 & 0 & 0 \end{array}\right].$$

*Consequently, $GH^{\prime }=\left[\begin{array}{cccc} 1 & 1 & \alpha & 0\\ 1 & 0 & 1+\alpha^{2} & 0\\ 0 & 1 & 0 & 1 \end{array}\right]$, and it has full rank 3, so the receiver can recover the source packets. Actually, in this case, the source packets can be recovered by merely GF(2)-based operations.*


In two special cases that $d_{t}=D$ and $d_{t}=0$, *i.e.*, receiver *t* has the highest and the lowest computational power respectively, (Equation 4) degenerates to a more concise form.

**Corollary** **2.**
*When $d_{t}=D$, (Equation 4) is equivalent to*

(8)
rank(GH)=n.

*When $d_{t}=0$, (Equation 4) is equivalent to*

(9)
$$\mathrm{rank}\left(H\right)-\mathrm{rank}\left(H_{d_{t}}\right)=n.$$



Recall that Fulcrum [13,14] can be regarded as a special RLNC scheme of our framework. One may notice that in Fulcrum, the decoding rule over GF(2) at a receiver is
(10)rank(H)=N,
which is sufficient but not necessary. In contrast, (Equation 9) is both necessary and sufficient. As to be seen in Section 5, there is an observable performance gain when (Equation 9) is adopted as the decoding rule instead of (Equation 10). Moreover, our proposed scalable RLNC is more flexible than Fulcrum, because the receivers with intermediate computational power can fully utilize its decoding capability to decode over intermediate fields (rather than only over GF(2)), so that the number of required coded packets can be reduced.

### 3.4. Decoding Complexity Analysis

In this subsection, we briefly analyze the computational complexity of the proposed scalable RLNC scheme at receiver *t* with the field constraint GF($2^{d_{t}}$). We assume that after a sufficiently large recoding process over GF(2), the last *r* positions in every received binary column vector h, which corresponds to the *r* precoded packets generated over the larger fields than GF(2), are nonzero. According to Corollary 1, when enough coded packets have been received such that the condition
$$\mathrm{rank}(G\left(HK_{d_{t}}\right))=n$$
is satisfied, receiver *t* can recover all original packets by linear combining *n* coded packets over GF($2^{d_{t}}$). Accordingly, it requires at most $n^{2}M/d_{t}$ multiplications and $n\left(n-1\right)M/d_{t}$ additions over GF($2^{d_{t}}$) in the decoding process. Following the same consideration in [4,18,19], we assume that it respectively takes $d_{t}$ and $2d_{t}^{2}$ binary operations to realize addition and multiplication between two elements in GF($2^{d_{t}}$). Consequently, the total number of required binary operations can be characterized as $O(Mnd_{t})$ to recover every *M*-bit original packet.

Herein, we did not consider the complexity to compute the inverse matrix of $GHK_{d_{t}}$ because in practice the packet length *M* is much larger than *n*, and this convention has also been adopted in [4,19] for computational complexity analysis.

## 4. Optimal Construction of Precoding Matrix G

Based on the analysis in the previous section, we are motivated to carefully design such a precoding matrix G that the full rank of H is equivalent to the full rank of GH, which can optimize the decodability performance for fixed parameters *n* and *N*. To achieve this goal, for the precoding matrix G, we first introduce the following condition that is stronger than the conventional maximal distance separable (MDS) property.

**Definition** **1.***An n×N matrix G over GF($2^{2^{D}}$) is said to be* MDS under GF(2)-mapping *if for any full-rank N×n matrix H over GF(2), rank(GH)=n.*

Recall that if G satisfies the conventional MDS property, all *n* columns in it are linearly independent. Obviously, the conventional MDS property is a prerequisite for the proposed MDS property under GF(2)-mapping. However, Example 3 demonstrates an MDS matrix G that is not MDS under GF(2)-mapping. To the best of our knowledge, except for a brief attempt in [13], there is no prior literature involving the construction of a matrix satisfying the MDS property under GF(2)-mapping. We next characterize an equivalent condition on the MDS property under GF(2)-mapping, so as to facilitate the explicit construction. Given an n×N matrix G, let C denote the set of row vectors generated by G:(11)$$C=\{mG:m\in \mathrm{GF}{\left(2^{2^{D}}\right)}^{n}\}.$$
For every c∈C, let $N_{c}$ denote its null space in $\mathrm{GF}{\left(2^{2^{D}}\right)}^{N}$.

**Theorem** **2.**
*An n×N matrix G is MDS under GF(2)-mapping if and only if*

(12)
$$\dim(N_{c}\cap \mathrm{GF}{\left(2\right)}^{N})<n,\;\forall \;c\in C\backslash \left\{0\right\}$$



**Proof.** We prove the theorem in a contrapositive argument. Assume that there exists a nonzero c∈C such that $\dim(N_{c}\cap \mathrm{GF}{\left(2\right)}^{N})\geq n$, and let m be a row vector over GF($2^{2^{D}}$) satisfying c=mG. Then, we can select *n* linearly independent column vectors $h_{1},\ldots ,h_{n}$ over GF(2) from $N_{c}$. Write $H=[h_{1}\ldots h_{n}]$. Thus, mGH=cH=0, so that GH is not full rank *n*, *i.e.*, G is not MDS under GF(2)-mapping.Assume that G is not MDS under GF(2)-mapping, and let H be a full rank N×n matrix over GF(2) subject to rank(GH)<n. Then, there exists an *n*-dimensional row vector m such that mGH=0. Write c=mG, so that cH=0. Since H is full rank *n*, there are at least *n* linearly independent vectors (which are the columns of H) belonging to $N_{c}$, *i.e.*, $\dim(N_{c}\cap \mathrm{GF}{\left(2\right)}^{N})\geq n$. ☐

For c∈C, let η(c) denote the number of elements in c belonging to $\mathrm{GF}\left(2^{2^{D}}\right)\backslash \left\{0,1\right\}$, and define an indicator δ which is set to 1 if c consists of an element equal to 1 and set to 0 otherwise. The following is a useful corollary of Theorem 2.

**Corollary** **3.**
*If an n×N matrix G is MDS under GF(2)-mapping, then the followings hold*

(13)
η(c)+δ>N−n,∀c∈C\{0}.


(14)
$$C\cap \mathrm{GF}{\left(2\right)}^{N}=\left\{0\right\}.$$



**Proof.** Assume there is a nonzero c∈C with η(c)+δ≤N−n, *i.e.*, N−η(c)≥n+δ. Define a new vector $c^{\prime }$ by restricting to its components belonging to GF(2), so that the dimension of $c^{\prime }$ is N−η(c). Thus, the dimension of the null space of $c^{\prime }$ in $\mathrm{GF}{\left(2\right)}^{N-\eta (c)}$ is N−η(c)−δ, which is no smaller than *n*. Correspondingly, $\dim(N_{c}\cap \mathrm{GF}{\left(2\right)}^{N})\geq n$, a contradiction to the MDS property under GF(2)-mapping for G according to (Equation 12).If there is a nonzero c∈C belonging to $\mathrm{GF}{\left(2\right)}^{N}$, then η(c)=0 so that (Equation 13) cannot hold as N>n, and thus G cannot be MDS under GF(2)-mapping. ☐ 

Conditions (Equation 13) and (Equation 14) are insufficient for the MDS property under GF(2)-mapping. The key reason is the possibility of the following
(15)$$\sum_{<j>}\alpha_{j}\in \mathrm{GF}\left(2\right),\;\alpha_{j}\in \mathrm{GF}\left(2^{2^{D}}\right)\backslash \left\{0,1\right\}.$$
For this reason, we should pay more attention in the matrix design to avoid the involvement of those elements in (Equation 15). The special case N=n+1 is easier to manipulate.

**Proposition** **1.**
*When N=n+1, an n×N matrix G is MDS under GF(2)-mapping if and only if (Equation 14) holds.*


**Proof.** The necessity has been shown in Corollary 3. To prove sufficiency, assume (Equation 14) holds for C defined in (Equation 11) based on G. Let c be an arbitrary vector in C. As (Equation 14) holds, η(c)>0. In the case η(c)=1, there must be at least one element in c equal to 1, because otherwise we can find another vector in C with all elements in GF(2), a contradiction to (Equation 14). Thus, $\dim(N_{c}\cap \mathrm{GF}{\left(2\right)}^{N})<n$ for this case. Consider the case η(c)≥2. Without loss of generality, write $c=[c_{1}\;\ldots \;c_{\eta (c)}\;0\;\ldots \;0]$ with $c_{j}\neq 0$. We can assume $c_{j}$ not all identical, because otherwise we can again find another vector in C with all elements in GF(2), a contradiction to (Equation 14). Moreover, for arbitrary two elements $a,b\in \mathrm{GF}(2^{2^{D}})$, a+b=0 if and only if a=b. Hence, there are at most η(c)−2 linearly independent vectors in $\mathrm{GF}{\left(2\right)}^{\eta (c)}$ that are in the null space of c, which further implies $\dim(N_{c}\cap \mathrm{GF}{\left(2\right)}^{N})<n$. We have proved (Equation 12) and thus the considered G is MDS under GF(2)-mapping. ☐ 

**Corollary** **4.**
*When N=n+1, there exists a systematic n×N matrix $G=[I_{n}\;A_{D}]$ over GF($2^{2^{D}}$) that is MDS under GF(2)-mapping if and only if $n<2^{D}$.*


**Proof.** Assume $n<2^{D}$. Define an *n*-dimensional column vector $a={\left[\alpha ,\alpha^{2},\ldots ,\alpha^{n}\right]}^{T}$, where α is a primitive element of GF($2^{2^{D}}$). In this way, all elements in a are distinct and every GF(2)-combination $\sum_{1\leq j\leq n}a_{j}\alpha^{j}$ among them does not belong to GF(2). By Proposition 1, $\left[I_{n}\;a\right]$ is an MDS matrix under GF(2)-mapping. When $n\geq 2^{D}$, let $a={\left[\alpha_{1},\ldots ,\alpha_{n}\right]}^{T}$ be an arbitrary *n*-dimensional vector in GF($2^{2^{D}}$). In order to make $\left[I_{n}\;a\right]$ MDS under GF(2)-mapping, according to (Equation 14) in Corollary 3, there is not any element $\alpha_{j}$ belonging to GF(2). If there is a basis, say $\left\{\alpha_{1},\ldots ,\alpha_{2^{D}}\right\}$ of GF($2^{2^{D}}$) in a, then 1 can be written as a GF(2)-linear combination of the basis, so that (Equation 14) does not hold. If there is not a basis of GF($2^{2^{D}}$) in a, then there exists an *n*-dimensional nonzero row vector v over GF(2) subject to va=0, so that (Equation 14) does not hold either. Thus, it is impossible for $\left[I_{n}\;a\right]$ to be MDS under GF(2)-mapping. ☐ 

Based on the above corollary, the required field size is exponentially larger than *N* in the construction of an n×N systematic MDS matrix under GF(2)-mapping. This implies that it is infeasible to construct such a practical precoding matrix G for large *N*. For this reason, it is alternative to choose to randomly generate G, which may cause a near-optimal decodability behavior as illustrated in the next example.

**Example** **4.**
*Define the following vectors $a_{1}={\left[\alpha \;\alpha^{2}\;\alpha^{3}\;\ldots \;\alpha^{7}\right]}^{T}$ and $a_{2}={\left[\alpha^{2}\;\alpha^{4}\;\alpha^{6}\;\ldots \;\alpha^{14}\right]}^{T}$ over GF($2^{8}$) in which α is a primitive element. It can be checked that both matrices $\left[I_{7}\;a_{1}\right]$ and $\left[I_{7}\;a_{2}\right]$ are MDS under GF(2)-mapping. Although the 7×9 matrix $G=[I_{7}\;a_{1}\;a_{2}]$ is not MDS under GF(2)-mapping, among 42435 7-dimensional subspaces of $\mathrm{GF}{\left(2\right)}^{9}$, there are only 127 instances to break the desired MDS property, that is, every basis for each of the instances forms a 9×7 matrix H with rank(GH)<7.*


## 5. Numerical Analysis

In this section, we numerically analyze the performance of applying the proposed systematic scalable RLNC scheme to a wireless broadcast network, which is a classical model to demonstrate the advantage of RLNC [4,5,6,7]. The number *n* of original packets in a batch is varied from n=6 to 24. In every timeslot, the source broadcasts one packet to all receivers. The memoryless and independent packet loss probability for every receiver is $p_{e}=0.2$, that is, in every timeslot, every receiver can successfully receive a packet with probability $1-p_{e}$. We consider the scheme with parameters D=2, r=2 where $r_{1}=r_{2}=1$. In the n×N precoding matrix $G=[I_{n}\;A_{1}\;A_{2}]$, the entries in $A_{1}$ and $A_{2}$ are randomly selected from GF($2^{2}$) and GF($2^{4}$), respectively. In the numerical analysis of scalable RLNC, the single source *s* has *n* original packets to be broadcast to a total of 30 receivers with different decoding capabilities. Specifically, the 30 receivers fall into 3 different groups and the 10 receivers in every group has the same decoding capability, and can decode based on the decoding rule (Equation 4) over GF(2), GF($2^{2}$) and GF($2^{4}$), respectively. In the first *n* timeslots, the source broadcasts *n* original packets, whose coding vectors are (n+r)-dimensional unit vectors, to all receivers. Starting from timeslot n+1, the source broadcasts coded packets, each of which is generated based on a random *N*-dimensional column vector h over GF(2), till all the receivers can recover the *n* original packets. Herein, for every parameter setting and every considered RLNC scheme, we conduct 1200 independent rounds of simulation which result in 95% confidence intervals.

Figure 2 depicts the average group completion delay per packet for the 3 groups of receivers, respectively labeled as “Scalable-GF($2^{x}$)”, x∈{1,2,4} of the considered scalable RLNC scheme. The group completion delay means the number of extra coded packets the source broadcasts till all the 10 receivers in the group can recover *n* original packets. For a better comparison, the figure also depicts the average group completion delay per packet, labeled as “RLNC-GF($2^{x}$)”, for a group of 10 receivers of three *different* classical systematic RLNC schemes over different fields GF($2^{x}$), x∈{1,2,4}. Recall that in the classical systematic RLNC scheme over GF($2^{x}$), the source first broadcasts *n* original packets and then randomly coded packets with *n*-dimensional coding vectors over GF($2^{x}$). One may observe from Figure 2 that for the case of GF($2^{4}$), the average completion delay of scalable RLNC is almost same as the classical RLNC. Over other smaller fields, even though scalable RLNC yields higher average completion delay than classical RLNC, it simultaneously guarantees the decoding compatibility at heterogeneous receivers, which cannot be endowed by classical RLNC schemes. For instance, assume that the source adopts classical RLNC over GF($2^{2}$) to generate coded packets. On one hand, the group of receivers with decoding capability constrained to GF(2) will fail to recover the original packets. On the other hand, the groups of receivers with decoding capability over GF($2^{4}$) cannot fully utilize their higher computational power so that the average completion delay cannot be further reduced compared with decoding over GF($2^{2}$). As a result, the performance loss for the cases of smaller fields in our proposed scalable RLNC compared to classical RLNC is the cost of decoding compatibility over different fields.

For the considered systematic scalable RLNC scheme, recall that for decoding over GF(2) in the proposed scalable RLNC, Equation (Equation 9) obtained in Sec. III is a necessary and sufficient rule while Equation (Equation 10), originally adopted in [13,14] for Fulcrum decoding, is a non-necessary rule. Figure 3 compares the average group completion delay per packet for 10 receivers as well as the average completion delay per packet at a single receiver when the receivers adopt different decoding rules (Equation 9) and (Equation 10) over GF(2). For the average completion delay at a single receiver, a noticeable performance gain can be observed. In particular, when the number of original packets is less than 10, the average completion delay at a single receiver is reduced by more than 20% based on the decoding rule (Equation 9) instead of (Equation 10). For the average group completion delay, the performance gain by adopting (Equation 9) instead of (Equation 10) becomes less obvious because it is offset by the increasing number of receivers in a group. Compared with Fulcrum, which only supports decoding over the smallest field GF(2) or the largest field GF($2^{2^{D}}$), in addition to the performance gain illustrated in Figure 3, our proposed scalable RLNC is more flexible. This is because the receivers with intermediate computational power can fully utilize its decoding capability to decode over intermediate fields (rather than only over GF(2)), so that the average completion delay can be reduced.

In the remaining part of this section, we shall analyze the performance of our scalable RLNC scheme by adjusting the *sparsity* $0<P_{h}<1$ of h, which controls to the probability for every component in h to be one. Specifically, for every packet to be transmitted by the source, the expected number of precoded packets to form it is $P_{h}\left(n+r\right)$. In previous analysis of this section, $P_{h}$ is set to 1/2. We next consider a more *sparse*
h with $P_{h}\leq 1/2$.

According to the work in [20], given that there are (i−1)(n+r)-dimensional linearly independent binary vectors with sparsity $P_{h}$, the probability that a new randomly generated (n+r)-dimensional binary coding vector $h^{i}$ with sparsity $P_{h}$ is linearly independent with them is lower bounded by
(16)$$1-{\left(1-P_{h}\right)}^{n+r-i}.$$

This bound indicates that except for the case *i* close to (n+r), the lower bound keeps very close to 1. Further, at the end of Sec. IV, we have illustrated that a random G will bring a near-optimal decodability behavior, that is, the full rank of H will lead to the full rank of GH with high probability. As a result, although our proposed scalable RLNC scheme with two-stage encoding process is different from the conventional sparse RLNC described in [20], we are motivated to bring the *sparsity* into our proposed scheme and attempt to meet a balance between completion delay and decoding complexity. The work in [14] has taken the sparsity into consideration in their performance analysis of Fulcrum, which is a special instance of our proposed scalable RLNC scheme.

In simulation, besides the consideration of sparsity $P_{h}$, we also extend the value range of *n* from [6,24] to [8,64] and set $r_{1}=r_{2}=2$. All other parameter settings are same as those in Figure 2. The 3 solid curves in Figure 4 illustrate the average group completion delay per packet for the 3 groups of 10 receivers under different field constraints GF(2), GF($2^{2}$) and GF($2^{4}$) for scalable RLNC with sparsity $P_{h}=1/2$. The 3 dotted curves in Figure 4 illustrate the completion delay performance under different field constraints GF(2), GF($2^{2}$) and GF($2^{4}$) for scalable RLNC with $P_{h}=1/4$. It is interesting to observe that with the batch size *n* increasing, under the same decoding constraint (*i.e.*, two curves in the same color), the completion delay performance for the case $P_{h}=1/4$ will converge to the case $P_{h}=1/2$. This result indicates that the lower bound in (Equation 16) is rather loose when *i* is close to n+r, and moreover, for a large enough batch size *n*, a more sparse vector h will not affect the completion delay performance much in a wireless broadcast network.

## 6. Conclusions

In this work, the proposed scalable RLNC framework based on embedded fields aims at endowing heterogeneous receivers with different decoding capabilities in complex network environments. In this framework, we derive a general decodability condition by the arithmetic compatibility of embedded fields. Moreover, we theoretically study the specific construction of an optimal precoding matrix G and illustrate the rationality of the near-optimal behavior of a randomly generated G.

In numerical analysis, we demonstrate that the proposed scalable RLNC not only guarantees a better decoding compatibility compared with classical RLNC, but also provides a better decoding performance over GF(2) in terms of smaller average completion delay compared with Fulcrum. In addition, the numerical analysis also demonstrates that for a large enough batch size, the sparsity of the vector h does not affect the completion delay performance much. As a potential future work, the theoretical insight behind this observation deserves a further investigation so as to facilitate the design of a scalable RLNC scheme with a better tradeoff between decoding complexity and completion delay.

Last, the present scalable RLNC framework assumes block-based coding. It would also be interesting to make use of the embedded fields structure to generalize the design of sliding window-based random linear coding schemes such as the ones studied in [21,22,23].

## Figures and Tables

**Figure 1 entropy-24-01510-f001:**
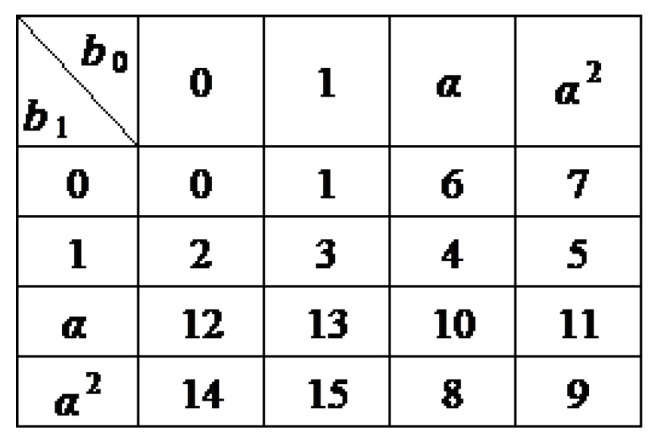
Every element $a_{0}+a_{1}\beta +a_{2}\beta^{2}+a_{3}\beta^{3},a_{j}\in \left\{0,1\right\}$ in GF($2^{4}$) has a unique expression in the form of $b_{0}+b_{1}\beta$, $b_{0},b_{1}\in \left\{0,1,\alpha ,\alpha^{2}\right\}=\mathrm{GF}\left(4\right)$, where $\alpha^{2}+\alpha +1=\beta^{2}+\beta +\alpha =\beta^{4}+\beta +1=0$. The integers 0 to 15 represent the decimal expression of the binary 4-tuple $\left(a_{3},a_{2},a_{1},a_{0}\right)$.

**Figure 2 entropy-24-01510-f002:**
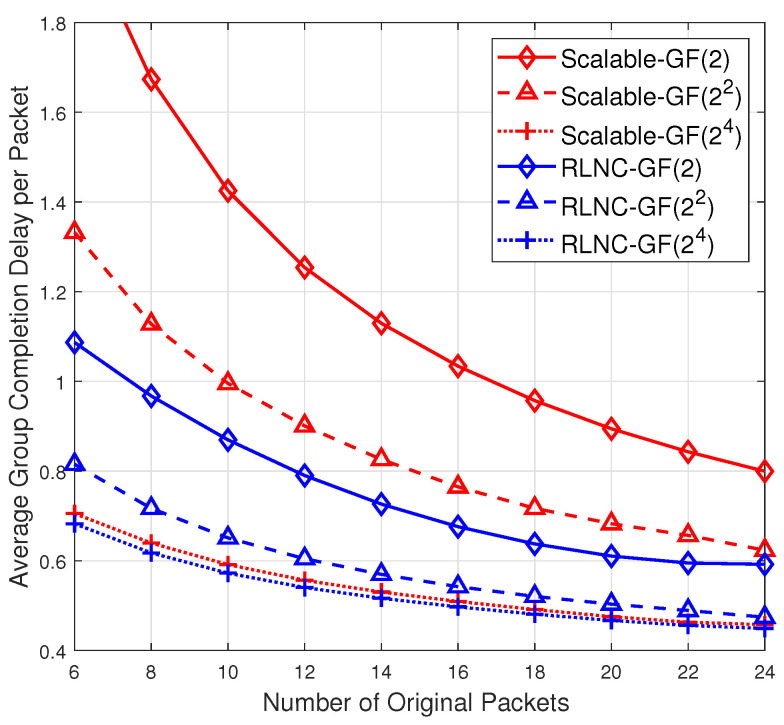
The average group completion delay per packet for the receivers of different systematic RLNC schemes in a wireless broadcast network with $r_{1}=r_{2}=1$ and packet loss probability $p_{e}=0.2$.

**Figure 3 entropy-24-01510-f003:**
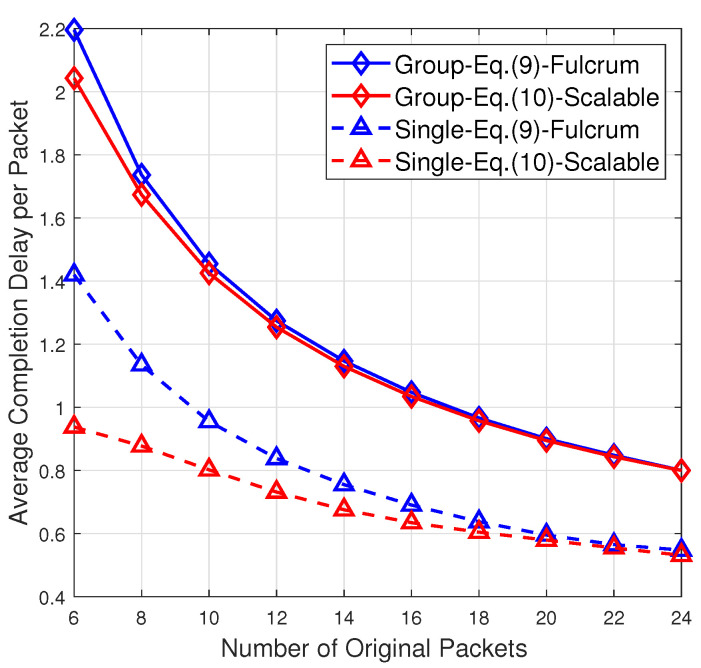
The average group completion delay per packet for 10 receivers as well as the average completion delay per packet at a single receiver when the receivers adopt different decoding rules (Equation 9) and (Equation 10) over GF(2).

**Figure 4 entropy-24-01510-f004:**
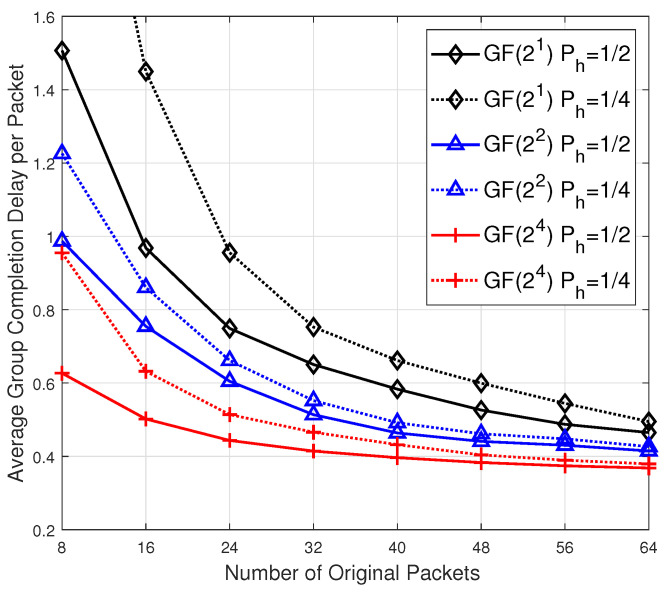
The average group completion delay per packet for scalable RLNC with different sparsity $P_{h}$.

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
