# Peer review of "Scalable Network Coding for Heterogeneous Devices over Embedded Fields"

_entropy, 2022, doi:10.3390/e24111510_

Round 1
Reviewer 1 Report
This manuscript presents a generalization of the multi-layer RLNC coding paradigm. The focus is on a generalized and optimized precoding that is implemented before the GF(2) coding. The encoding and decoding principles are mathematically analyzed in detail. Further, the optimization of the construction of the precoding matrix is examined rigorously. Illustrative numerical results for a classical wireless broadcast setting demonstrate the strengths of the proposed generalized approach.
It would be helpful for the reader to briefly contrast the proposed approach from other multi-layer RLNC coding schemes, e.g., Tang, et al. "Scalable Network Coding over Embedded Fields." Proc. IEEE/CIC International Conference on Communications in China (ICCC), 2021; Yazdani, et al "Revolving codes: overhead and computational complexity analysis." IEEE Communications Letters, 2020.
The encoding and recoding steps, as well as the different cases of the decoding are analyzed in good detail. Can the algorithmic complexity of the decoding be characterized formally?
The double power requirement of the column vector results in fairly large “step sizes” of the packet sizes. Padding is mentioned as a solution, but with these large step sizes a lot of padding may be needed. Would approaches that avoid padding, e.g., Schütz, "Packet-Preserving Network Coding Schemes for Padding Overhead Reduction." Proc. IEEE LCN, 2019; Taghouti, et al. "Reduction of padding overhead for RLNC media distribution with variable size packets." IEEE Trans. Broadcasting, 2019 be compatible with the proposed approach?
The proposed approach focuses on block-based coding, which is fine. It would be interesting though to briefly clarify whether and if so how, the proposed approach could be adapted to window-based RLNC, such as approaches Karetsi, et al., "Lightweight network-coded ARQ: An approach for Ultra-Reliable Low Latency Communication." Computer Communications, 2022; Ma, et al. "Sliding-Window Based Batch Forwarding using Intra-Flow Random Linear Network Coding." Proc. IEEE IWCMC, 2020; Tasdemir, et al. "FSW: Fulcrum sliding window coding for low-latency communication." IEEE Access, 2022. Could such a combination of the proposed approach and a window based approach even further reduce the delays?
The numerical work is interesting and insightful, but relatively sparse for this paper. This is probably ok for this first paper on the generalized multi-layer coding. However, please clarify the evaluation settings and parameters in more detail. Were the evaluation results obtained with discrete event simulations? If so, what is the number of independent replications? What are the resulting statistical confidence intervals?
Minor correction:
Line 119, “by by” - > “by”
Overall, this is a very interesting manuscript on an important topic in reliable wireless networking. Heterogeneous wireless devices are very common. However, the support for heterogeneous wireless devices is largely lacking so far. This manuscript makes an important step towards fundamentally advancing the coding in support of heterogeneous wireless devices. This is a difficult area of study and this paper addresses this challenging topic area in a rigorous manner and makes very good scientific progress. Therefore, eventual acceptance is recommended subject to the moderate revisions/clarifications outlined above.
Author Response
See attached file. "Response to reviewer1_ entropy-1971183"

Reviewer 2 Report
This paper proposes a scalable random linear network coding framework based on embedded fields to endow heterogeneous receivers with different decoding capabilities. The numerical analysis illustrates that the proposed scheme guarantees a better decoding compatibility over different fields compared with classical over a single field, and outperforms Fulcrum in terms of a better decoding performance. In my opinion, this work is meaningful for network. There are some comments below which I recommend to give one chance to take a revision. A more comprehensive literature survey may be provided with multi-associated parameters aggregation-based routing and resources allocation in multi-core elastic optical networks. The simulation should add in more details to make the proposal clearly.
Author Response
See attached file. "Response to reviewer2_ entropy-197118"
